# Shipwrecked on the Rock, or Not Quite: Gypsophytes and Edaphic Islands

**DOI:** 10.3390/plants13070970

**Published:** 2024-03-27

**Authors:** Juan F. Mota, Fabián Martínez-Hernández, Francisco Javier Pérez-García, Antonio Jesús Mendoza-Fernández, Esteban Salmerón-Sánchez, M. Encarna Merlo

**Affiliations:** 1Department of Biology and Geology, University of Almería, 04120 Almería, Spain; jmota@ual.es (J.F.M.); fpgarcia@ual.es (F.J.P.-G.); esanchez@ual.es (E.S.-S.); emerlo@ual.es (M.E.M.); 2Department of Botany, University of Granada, 18071 Granada, Spain; amf788@ugr.es

**Keywords:** gypsicolous, gypsophile, gypsophily, gypsum outcrops, gypsovag, species–area relationships (SAR)

## Abstract

Species–area relationships (SAR) constitute a key aspect of ecological theory and are integral to other scientific disciplines, such as biogeography, which have played a crucial role in advancing biology. The theory of insular biogeography provides a clear example. This theory initially expanded from true islands to other types of systems characterized by their insularity. One such approach was linked to geoedaphic islands, as seen in gypsum outcrops. While these continental areas have been considered insular systems, only limited and mostly indirect evidence thereof has been provided. This study utilized SAR to advance the understanding of gypsum outcrops as insular continental territories. It is hereby hypothesized that gypsum outcrops are edaphic islands, although their insular nature depends on the different functional or ecological plant types, and this nature will be reflected in the potential Arrhenius model *z* values. The results obtained support both hypotheses and provide insight into the ecological factors that help interpret the insularity of these areas. This interpretation goes beyond their mere extent and the distance among outcrops, emphasizing the importance of environmental filters. Said filters vary in permeability depending on the degree of gypsophily, or preference for gypsum, exhibited by different species.

## 1. Introduction

It is undeniable that the theory of evolution was largely forged in island settings. Both Darwin and Wallace made insightful observations in these systems that contributed to the formulation of their theory, which unifies the entire realm of biodiversity [1]. However, this is not the only biological theory born from island observations. The Theory of Island Biogeography (TIB) by MacArthur and Wilson [2] marked another qualitative leap in this field of research by establishing that species’ diversity on an island is determined by its rates of colonization and extinction, as well as its distance from the mainland (or source of propagules) and its size. Regarding the latter factor, MacArthur and Wilson [2] noted that theories like island biogeography are often reached using stepping stones, and the species–area curves are such stepping stones. The idea of size as a factor to explain the species richness of a given territory was previously born relying on the species accumulation curves (SAC). The concept of SAC antecedes that of ISAR (island species–area relationships) [3]. SAC have been employed for a considerable period of time to delineate how the number of species increases with the size of the sampled area [4,5,6,7] or the area of various islands in an archipelago [8,9,10]. According to Gleason [11], this concept was initially formalized by Arrhenius [12] in a mathematical model with the expression *S = c·A^z^*, where *S* represents the number of species, *A* is the area, *c* is a coefficient, and *z* is the slope of the model. Arrhenius asserted that the results he obtained, spanning various types of plant communities, demonstrated the validity of the formula across areas of different sizes, ranging from square decimeters to hectares. According to García Martín and Goldenfeld [13], the area–species relationship is a robust consequence of a species’ abundance distribution skewed towards the rarer species, resembling a log-normal distribution, and has been demonstrated across a wide variety of organisms and climates.

However, the relationships between species and areas were well known even before the formulation of those equations by Arrhenius and Gleason [14]. This topic has been widely debated and is considered by many as one of the few principles in ecology [10,15,16]. In both the Arrhenius and Gleason models, the intersection (*c*) quantifies α-diversity, while the slope (*z*) measures β-diversity, providing an assessment of the differentiation between habitats or habitat fragments. The constant *c* can be interpreted as the number of species expected to be found in a unit area, with its value depending on the exponent *z* and the spatial scale under consideration [13]. The parameter *z* also represents an estimate of the degree of insularity of fragmented or disjointed territories [17,18], as observed in gypsum outcrops and other geoedaphic islands [19,20,21,22]. In general, higher values of *z* are interpreted as indicating higher isolation; thus, a low slope indicates less sensitivity to island area compared to a high slope [23]. It is essential to consider that the values of *z* not only respond to geographic distance but can also vary based on a series of system properties and specific biological processes [2,10,23,24,25]. This value depends on the habitat, scale, and taxa under consideration in the analyses [26]. Thus, *z* values in the range of 0.1–0.4 are common for plants and birds in island groups, while values close to unity are common for intercontinental scales. In the case of microbes, organisms with an extraordinary dispersal capacity due to their small size, values as small as 0.05 are typical [13].

Von Humboldt, Darwin, Wallace, and other early naturalists observed that insularity manifested itself on both oceanic and offshore (continental) islands as well as on mainland islands [18]. Island biogeography has been extended to include other insular habitats that are not necessarily surrounded by water and can refer to any patchy habitat in a hostile or contrasting matrix [27,28,29]. This prediction of island biogeography has found strong support in data [16]. Edaphic island systems differ from other types of island ecosystems in that they are characterized by patches of distinct soil conditions supporting specific vegetation types, which are distinct from the surrounding landscape [30,31]. For this reason, some authors [17,32] have cautioned that ISAR and SAC are not equivalent. In this regard, Mendez-Castro et al. [18] note that the impact of insularity on terrestrial systems resembling islands is conceptually and methodologically challenging because recognizing species’ sources and measuring isolation is not as straightforward as it would be on true islands. Kruckeberg [20] places the first specific studies of habitat islands on the mainland in the 1970s, focusing on fauna. Although several studies preceding those mentioned had addressed species–area relationships (SAR) in plants [33,34], there was no specific investigation of SAR in continental island-like systems until the subsequent decades; however, none of them were dedicated to gypsum.

Gypsum outcrops have been considered as edaphic islands in numerous studies [35,36,37,38,39,40]. However, in almost all of these studies, their insular nature has been assumed a priori without direct evidence to support it. From this point onward, the present study’s main hypothesis (H) follows this line of reasoning and posits that gypsum outcrops function as edaphic islands for plants (H1), although not uniformly for the different functional types that thrive there (H2). This second hypothesis is expanded to the notion that, for species adapted to gypsum, the gypsophytes, the insular effect of these outcrops is less pronounced than for indifferent or generalist species (gypsovags). Both hypotheses (H1 and H2) conflict with one of the most widespread assertions regarding the dispersal capacity of gypsophile species, which has been assumed to be predominant at short distances [36,41,42,43,44]. On the other hand, an expanded version of the second hypothesis would support the observation made by Izco [45] regarding the remarkable ability of these plants to trace the territory with their seeds and colonize suitable environments, even those distant from their normal populations, where they competitively thrive against other vegetation.

In this way, the present study introduces an original approach for three key reasons: 1.- The insular nature of gypsum outcrops based on the slopes (*z*) of SAR curves is addressed; 2.- It is proposed that the differential insular nature of these outcrops varies according to the degree of gypsophily or preference for gypsisols exhibited by the species (gypsophytes vs gypsovags); 3.- It is analyzed whether or not the degree of stenochory or endemicity of these species (wide-area gypsophytes vs narrow-area gypsophytes) is related to the insular nature of these outcrops.

In relation to the hypotheses raised and the three points expressed earlier, several functional groups of plants were distinguished (according to their greater or lesser degree of gypsophily), in two taxonomic groups at the family level (*Poaceae*/*Asteraceae*), according to their ecological preferences (ruderal of Mediterranean shrublands). Ten randomly generated groups were added to previous ones. In all cases, efforts were made to ensure that the number of species was approximately equal to that of the reference groups, which included species with gypsum preferences, i.e., gypsophiles and gypsoclines. This avoided bias when comparing the parameter *z* values obtained from the SAR curves obtained for each of these groups. Based on the hypotheses and methodology outlined, this study’s approach is closely related to that applied to vascular plant diversity in other island systems [46].

## 2. Results

The model implemented to construct species accumulation curves for the various studied outcrops exhibited a satisfactory fit, with the more extensive ones showcasing a higher number of species (Figure 1a). This same pattern was reiterated for the average number of species found in the 1000 m^2^ plots (Figure 1b). In both instances, the Sorbas outcrop (SO) displayed the highest richness, followed by Venta de los Yesos (VY).

Furthermore, the comparison of species’ richness in the 1000 m^2^ quadrats sampled across various outcrops, grouped into three categories based on the order of magnitude of their surface areas, revealed significant differences among outcrop size groups according to the species’ degree of gypsophily, as presented in Table 1. These same results were also obtained for the *Poaceae* and species of the *Gypsophiletalia* order. For *Asteraceae* and species of the *Stellarietea mediae* order, the results differed exclusively, while for groups 1 and 2, no significant differences (s.d.) were found. In the case of the *Rosmarinetea officinalis* species, only s.d. were found between group size 2 and 3. In summary, out of the 24 group comparisons, 20 recorded s.d.

Additionally, in all cases, significant differences (s.d.) were found for this parameter between the group of gypsophytes (Gy) and gypsophytes + gypsoclines (GyGc) and the rest of the predefined species sets, whether considering taxonomic groups (*Poaceae* and *Asteraceae* families), syntaxonomic groups (*Stellarietea mediae*, *Rosmarinetea officinalis*), or functional groups (e.g., gypsovags), as well as those established randomly. These differences were observed not only in direct comparisons (*p* > 0.001) but also in the values obtained for the rank–biserial correlation (r_B_), which, in all cases, were large (Table 2 and Table 3).

Figure 2 visually summarizes the results of the non-parametric Wilcoxon’s paired samples test conducted, the details of which are reflected in Table 2 and Table 3. In summary, it can be stated that the slopes obtained for the group of gypsophytes (Gy) and the group of gypsophytes + gypsoclines (GyGc) exhibited values lower than those observed for the remaining groups under consideration.

In all instances, the disparity in parameters generated by the model for groups incorporating gypsophile species, or preferably gypsophiles, is evident when compared to the rest. However, these differences are particularly pronounced with what could be considered their complementary groups, i.e., those comprising indifferent or gypsovag species (Ros, GV, or P_nGR). In these cases, the *z*-values are considerably higher than those exhibited by gypsophile species (Figure 2).

It is noteworthy as well that, for these same groups, *c* values were higher than those of the remaining groups (see Table 2), excluding those encompassing the entirety of the plot species (W) or the group of gypsovags (Gv). However, both contingents include a substantially greater number of species compared to those with gypsophile preferences (Table 2 and Table 3). Although not explicitly presented, the differences were nearly always statistically significant when compared to the remaining groups.

The comparison of *z*-values for wide-area gypsophytes (wG) and narrow-area gypsophytes (nG) did not yield any significant differences in the two-sample paired test (Table 4).

## 3. Discussion

Similarly to other studies considering the insular nature of special rock outcrops [27,47,48], the species–area relationships (SAR) provided a well-fitted model for the gypsum outcrops under investigation, encompassing their entire vascular flora (Figure 1a). While this study’s models indeed describe richness as a function of island size, it is important to note that this isolated result alone does not definitively establish the presence of an island effect. Similar outcomes could be anticipated solely based on the size of these outcrops. Nevertheless, applying this model to species’ richness in uniformly sized sample plots, in this case, 1000 m^2^, reveals an effect that extends beyond the outcrop size (Figure 1b). The data gathered in Table 1 align with the same perspective, as sample plots from larger outcrops exhibited a higher number of species compared to those situated in smaller outcrops. Since the plot size remained constant, these data indicate an explanatory factor of an island effect or, to put it in other words, that the outcrops are differentially accessible. This circumstance implies recognizing constraints in relation to the surrounding land matrix. Most previous research on such geoedaphic islands refers to a hostile matrix [49,50]. However, maintaining this argument becomes challenging in the case of gypsum outcrops, given that over 90% of the hosted flora can be considered gypsovag, meaning that, a priori, the mentioned matrix is permeable to a high percentage of species. This leads to the conclusion that, for most of these species, the distance to the source of propagules or immigrants should be considered as null or nearly null. In this case, identifying the putative source of migrants [18] becomes challenging, although, in terms of size, the role could be attributed to the Sorbas outcrop. However, gypsophile species are also absent here, such as *Lepidium subulatum* and *Frankenia thymifolia*, widely distributed in Spanish gypsum outcrops and present in almost all other outcrops [35]. Other studies have reached similar conclusions regarding the effects of the surrounding matrix [51]. This aspect has been highlighted as a key point in interpreting the insularity of these island-like habitats [28]. However, the *z*-values obtained in this research fall within the range that can be attributed to island-like systems [27,52]. Applying the Arrhenius power model to the sampled plots yielded a mean value for the *z* exponent of 0.245 ± 0.036, encompassing all recorded species (Table 2). This value falls within the ranges considered normal, 0.15–0.35 [53], which also aligns with the more common values reported for plants and birds in island groups (0.1–0.4) by García Martín and Goldenfeld [13]. Furthermore, the indicated value approaches the upper end of these ranges, typical of island-like habitats [27,54] and very close to the “canonical” value proposed by Preston (0.265). However, the most noteworthy aspect of the collected values for *z* in Table 2 and Table 3 is not this, but rather the comparison between them based on the different groups considered. With the exception of groups including gypsophyte and gypsocline species, all *z*-values were above 0.20. Only for gypsophile and gypsocline species are these values lower than expected. This appears to be a paradox since, to the extent that these species are restricted to gypsum, their values should be equal to or higher than those for the rest of the groups, and it would be expected that distance would exert some influence.

A nuanced interpretation of these findings can be aided by considering other systems of a similar nature, such as high mountain environments [51,55]. Even in a strict sense, there is no separation distance between summit areas and the source of propagules in those cases, either, i.e., they primarily arrive from the immediate ground matrix below. However, it is evident that environmental filters exist, preventing some species from establishing and thriving in the summit zones. The well-established double gradient of temperature and precipitation, which regularly occurs with increasing altitude, is widely known [56]. In the case of high mountain environments, these factors and their associated effects could serve as a primary explanatory factor for the island-like nature of these habitats [57]. Similarly, for gypsum soils, a comparable explanation can be offered, as they represent an extreme habitat for vascular plants [58]. Numerous pieces of evidence support this hypothesis, including water scarcity [59,60], deficiencies in macronutrients [61,62], and challenges in seedling establishment, including those of invasive species [63]. These environmental filters may also be responsible for the spontaneous processes of plant autosuccession that occur in these outcrops following the dramatic disturbances caused by gypsum quarries [64].

The outstanding question revolves around why the distance factor does not exert a clear influence on gypsophile species, despite their absence from the surrounding areas of the outcrops due to their restriction to gypsum. Several factors could underlie this paradox. The first of these might involve attributing to these species a significant capacity for long-distance dispersal. Nevertheless, as previously noted, many authors have argued to the contrary. Furthermore, they have invoked the opposite argument to justify the restriction of these species to gypsum soils, relying on the absence of special adaptations in their seeds and diaspores [36,41,42,43]. In this, they align with generalizations made about plants specific to these geoedaphic island-like systems [31,65]. However, perhaps insufficient attention has been paid to the fact that in gypsum steppes, birds whose diet is largely based on seeds are common, as seeds indeed constitute a critical food source for fauna in these highly unproductive environments [66]. This idea is supported by the presence of seeds from both gypsophytes, *Ononis tridentata*, and *Gypsophila struthium*, in the crops of *Alectoris rufa* [35,67]. In the case of this latter species, it is documented that *Galerida theklae*, Thekla’s lark, feeds on it in the Hoya de Baza [68], a territory where gypsum shrub-steppe is very abundant. While precise data may be lacking for other gypsophytes, information is available for other members of some of the families of flowering plants well-represented in gypsum environments, such as *Brassicaceae* or *Caryophyllaceae* [69]. Although most steppe birds living in the Iberian Peninsula are not migratory, they may carry out shorter seasonal movements within their distribution range in response to changes in food availability or climatic conditions. These movements may involve displacements within their local habitat or towards adjacent areas with more suitable resources, among which seeds are key. A stepping-stone type of dispersal, even over long distances [70], could explain the wide distribution of many gypsophytes, along with their significant ability to establish themselves on gypsum.

Another possible interpretation is that, in some way, the surrounding matrix of gypsum outcrops is also highly permeable for gypsophytes, or at least, it was at some point in the past. Perhaps what is observed today represents echoes from the past, where gypsophytes took advantage of past climatic crises during which the hostile matrix experienced reduced competition and increased accessibility for gypsophytes. Some phylogeographic studies of gypsophilous flora seem to support this notion [71]. Some authors have hypothesized that insularity and harsh soil conditions favour enhanced plant persistence strategies [72]. After colonization, the unquestionable persistence ability of these plants becomes evident as they can utilize resources very conservatively, which is demonstrated by their ability to subsist with minimal levels of essential nutrients such as P [61]. Perhaps what is currently observed is nothing more than the outcome of the great persistence capacity of gypsophile species in these gypsum refuge environments [42].

Regarding the effect of outcrops on the slopes of wG as compared to nG, this study’s initial hypothesis attributed lower values to the former, assuming that their broader distribution could imply a greater dispersal capacity. The results have not substantiated this initial idea, although there is also no evidence to the contrary (Table 4). Revisiting this hypothesis, it is possible that wG may indeed have a greater dispersal capacity (e.g., *Ononis tridentata* and *Gypsophila struthium* [67]); however, nG have the advantage of local adaptation, i.e., they are adapted to the climatic conditions, characterized by maximum aridity in the southeastern Iberian Peninsula. This would imply that the non-gypsum matrix between outcrops might be less environmentally “hostile”, a circumstance that may not be the same for wG, especially if they are gypsum specialist plants. In this way, the non-gypsum matrix could pose a greater challenge to traverse. In fact, the distribution of nG is more compact, indicating that their area of occupation closely approximates their extent of presence compared to wG [73].

## 4. Materials and Methods

This research was conducted in six gypsum outcrops in the province of Almería, in the southeastern part of Spain, a hotspot of diversity for Spanish gypsophilous flora [35,74]. The locations of these outcrops are shown in Figure 3, and their main characteristics are detailed in Table 5. Additional details regarding the environmental characteristics of each of these outcrops can be found in Mota et al. [35]. Four of them are either entirely or partially included in the Spanish network of protected areas and integrated into the EU Natura 2000 network. However, their comprehensive conservation is not guaranteed, as the Natural Resources Management Plan (PORN, Plan de Ordenación de los Recursos Naturales) surprisingly considers mining extraction as a compatible activity with their conservation [75].

The main plant communities in the area, central to the research presented here, are classified in the EU Habitats Directive as priority habitats (Iberian gypsum vegetation, *Gypsophiletalia*; Garrigues occupying gypsum-rich soils of the Iberian; EU Habitats Directive, Annex I habitat type, code 1520). These are shrublands that inhabit gypsum-rich soils in the Iberian Peninsula, characterized by their sparse nature and the presence of numerous gypsophilous species. Among them are various species of the genus *Gypsophila* (e.g., *G. struthium*), *Helianthemum squamatum*, *Lepidium subulatum*, *Frankenia thymifolia*, *Reseda stricta*, and *Ononis tridentata*. In addition to these widely distributed species, several regional and local endemics in the study area include *Coris hispanica*, *Santolina viscosa*, *Chaenorrhinum grandiflorum*, *Teucrium turredanum*, or *Helianthemum alypoides* [35].

In each of these outcrops, five plots of 50 × 20 m were established with nested subplots, following the protocol outlined by Barnett and Stohlgren [80]. Details on the utilization of these plots and sampling for this habitat can be found in Mota et al. [64]. Within each of these plots and their corresponding subplots, the presence of all vascular plant species was recorded. The samplings were conducted in spring, all concentrated in the months of March, April, and May, as flowering occurs early in these territories, with the exception of some perennial gypsophytes, which are easily recognizable in their vegetative phase. A total of 186 species were identified during the study, encompassing both annuals and perennials.

Species–area relationships were modelled using the data collected at each site. Consistent with the Island Biogeography Theory [2], power-law regression was applied to derive slopes from log–log transformed plot area and species richness data. According to García Martín and Goldenfeld [13], among the various models relating area (*A*) to the number of supported species (*S*), the power-law equation (*S = c·A^z^*) is the most commonly utilized form and has been employed for diverse organisms [10,27].

Using this regression model, the number of species [81] was initially related to the surface area of each outcrop. The surface area was obtained using the method proposed by Ochoterena et al. [82] and employed two open-access versions of ArcGIS Landsat imagery focused on short-wave infrared spectra: Landsat 7 imagery with channels 7, 4, and 3, and Landsat 8 imagery with channels 7, 6, and 4; consequently, gypsum appeared turquoise in color under these channel combinations. The obtained area served as the independent variable for a significant portion of the conducted analyses. To ascertain if there were significant differences (ANOVA) in species richness, the outcrops were grouped into three categories based on their size. These categories can be interpreted as large, medium, and small (Table 5). With the exception of the data used to create Figure 1, the rest of the SAR analyses were based solely on the species recorded in the sampling plots. The set of species in each plot represents its floristic richness. From this set of species, subgroups were established based on various criteria, which almost always had a functional and ecological basis; taxonomic/syntaxonomic groups (families/phytosociological classes) were also employed, along with some mixed characteristics in certain cases (Table 6).

Given the gypsophilous nature of the soils, one criterion employed was the preference of plants for this type of substrate (gypsophily). Thus, plants were differentiated into gypsophiles, gypsoclines, and gypsovags [35,83,84]. While the former group of plants includes specialists exclusive to gypsum soils, the latter corresponds to the category indifferent to soil nature, being common outside gypsisols [35,62,84,85,86,87]. Among the gypsophytes, narrow-(nG) and wide-area gypsophytes (wG) can be distinguished according to their distribution area. The group of plants that exhibit a preference for gypsum soils without being exclusive to this substrate is referred to as gypsoclines [42,83,88,89,90]. Some of the species included in this category display many or all of the gypsum-related adaptations exhibited by those considered specialists, such as their ability to store Ca, Mg, and S, in addition to a certain degree of succulence [61]. For these reasons, they were included in some analyses alongside the gypsophytes. In reality, the number of species in this functional group (Table 5) does not allow for independent analyses, especially in the case of small outcrops.

Another criterion employed for species grouping was taxonomic, such that some species accumulation curves were exclusively constructed for the species of the two most abundant families in the samples, *Asteraceae* and *Poaceae*. SAR were also analyzed for species belonging to the phytosociological classes *Stellarietea mediae* and *Rosmarinetea officinalis*. In this latter class, gypsophile species of the order *Gypsophiletalia* were excluded as they are integrated within *Rosmarinetea officinalis* [91]. In considering these groups to compare the *z*-values obtained from the Arrhenius power function, our approach largely coincides with those proposed in other research on SAR [26,46]. In all cases, except when considering the entirety of the species recorded in the plots, the number of species included in the analyses approached that of the gypsophiles, the target group of this investigation. The decision to adopt this approach was grounded on the crucial premise that the execution of robust comparative analyses of the parameters *c* and *z* from the obtained curves requires the absence of substantial disparities among these values across the examined species sets [53].

To complement the analyses, and in accordance with the previously discussed criteria, 10 random groups of perennial species were generated for each plot, from which values for the parameter *z* were also computed. In this instance, the restriction to perennial species was motivated by various factors, such as the substantial proportion of gypsophytes being perennial, and considering that the presence of annual species can vary significantly based on seasonality and annual precipitation levels [64]. As previously indicated, in accordance with MacArthur and Wilson [2] and Preston [33,34], highly isolated communities exhibit a slope exceeding 0.2, whereas non-isolated communities display values below 0.17 (also refer to Connor and McCoy [24]). These and additional values documented in the references are mentioned here because they have been utilized as benchmark values in the discussion of the results.

The parameter *c*, which reflects the patterns of species’ richness in both outcrops and plots [2,23], is integral to the potential model. Its interpretation holds relevance in the analysis of the results, as previously noted [25].

To compare the *z* values (see Table 2 and Table 3), a non-parametric paired mean analysis was conducted with JASP (Version 0.18.3) [92]. This involved considering two columns of data. One of them always represented the group of gypsophytes or the one that included gypsophytes + gypsoclines. The second column of data corresponded to each of the groups reflected in Table 6, with the exception of wG and nG, and the 10 randomly created groups. After ensuring whether the differences between these pairs of samples followed normality (Shapiro–Wilk), it was found that, although this was the case in most comparisons (>80%), it was more appropriate to resort to the non-parametric Wilcoxon’s signed rank test. With the paired *t*-test, the null hypothesis (H0) is that the pairwise difference between the two groups is zero. The results of these analyses are provided in Table 2 and Table 3, which include the Hodges–Lehmann estimate (the median difference between the two groups) and the rank–biserial correlation (rB). This latter parameter can be considered as an effect size, and values < 0.1 are interpreted as “trivial”, “small” if they are 0.1–0.3, “medium” if they are 0.3–0.5. Above this last limit, they are considered “strong” [93].

In the case of wG and nG, the comparison of the obtained z-values was conducted using a *t*-test, supported by the Bayes factor, and a sign test. As these groups include a limited number of species, they were not considered in comparisons with other groups, although the *z*-values obtained for them fell within the range calculated for the overall group of all gypsophytes (see Table 4).

## 5. Conclusions

The flora of gypsum outcrops conforms to species–area relationship (SAR) patterns, with larger outcrops containing a greater number of species; the correlation between the two variables, using the power curve, is strong (R > 0.80). Utilizing the Arrhenius potential model for the sampled 1000 m^2^ plots and considering all species found, the value of *z* for these gypsum outcrops (*z* = 0.25) places them at the level of other continental island-like habitats. 

Among the different species groups considered in this study, the values of *z* are lower for the set of gypsophiles and gypsophiles + gypsoclines. This highlights that the outcrops have a differential effect, or in other words, they are more propitious for one type of species compared to others based on their degree of gypsophily. One possible interpretation is that for species adapted or pre-adapted to living on gypsum, eco-physiological barriers or filters are less pronounced than those for gypsovags. These latter species, although capable of living on gypsum, do not exhibit specific adaptations to it. As a result, gypsum outcrops have a higher degree of insularity for gypsovag species than for gypsophiles and gypsoclines. This phenomenon is attributed to the existence of environmental filters whose effects would be analogous, but not homologous, to the distance to the mainland on true islands. In other words, gypsum hinders the entry of non-adapted species (physiological barrier), requiring a larger quantity of propagules (attempts) to establish themselves there. The larger size of the outcrop would increase the number of propagules and species capable of reaching there, consequently increasing the number of attempts to establish themselves. The distance, in this case, would not play a significant role since the surrounding territories are separated by a distance of 0 for more than 90% of the flora populating them, which is gypsovag in nature. The key factor is the degree of tolerance or pre-adaptation to gypsum, which could be interpreted as a “pseudodistance” or “adaptation distance”.

Conserving these edaphic islands is essential if they indeed offer the kind of microrefugia demonstrated to mitigate the extinction risk posed by recent climate change. Furthermore, if they are completely destroyed, as can happen with mining or intensive agriculture, their recovery may be impossible under the current environmental conditions. This information can be useful for conservation efforts in island-like continental environments, as it highlights the importance of preserving the unique soil conditions and vegetation types found on edaphic islands. The conservation of species’ diversity on edaphic or soil islands requires the protection of specific habitats, and if, as in the case of gypsum, they are largely degraded, the restoration of areas that allow for the maintenance of species’ diversity is also necessary. In short, gypsum is essential for conserving biodiversity because without gypsum, there are no gypsophytes.

## Figures and Tables

**Figure 1 plants-13-00970-f001:**
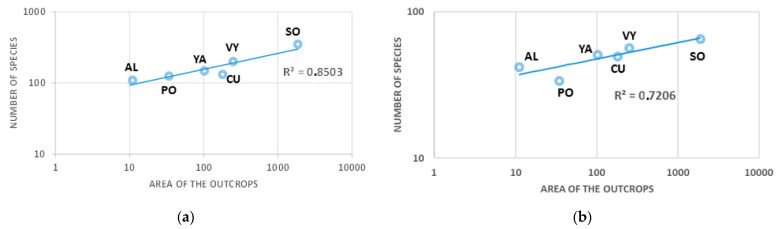
Species–area relationships (SAR) between the size of the studied outcrops (**a**) and the total number of vascular species, and (**b**) the average species richness of the sampled 1000 m^2^ plots. Abbreviations for the sites: Sorbas (SO), Venta de los Yesos (VY), Cueva de los Úbedas (CU), Alfaro (AL), Yesón Alto (YA), and Polopos (PO).

**Figure 2 plants-13-00970-f002:**
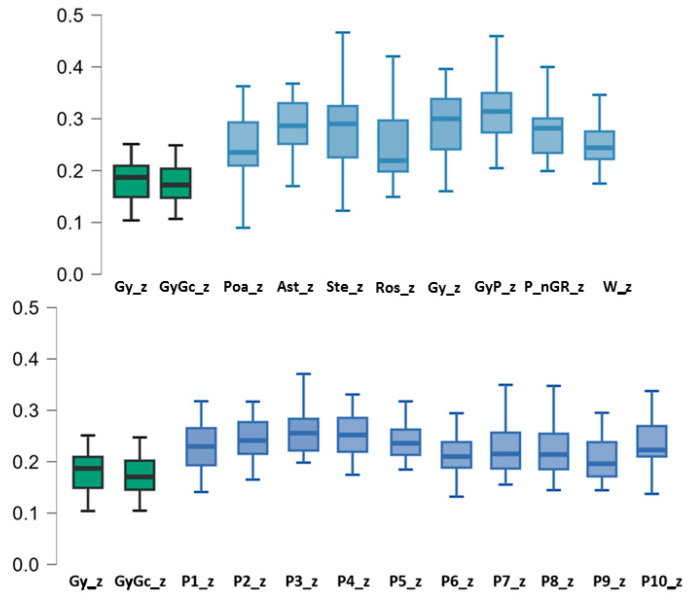
Box plots for the median *z*-values calculated for the various plant groups under consideration are depicted. Additionally, confidence intervals (95.0%) are presented.

**Figure 3 plants-13-00970-f003:**
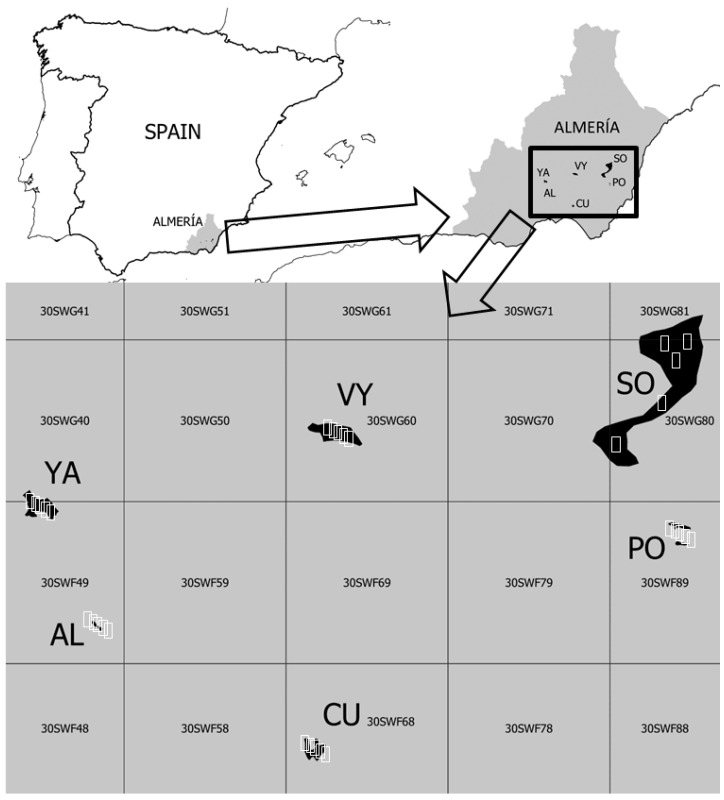
Location of the gypsum outcrops and sampling plots (white rectangles) studied. UTM (Universal Transverse Mercator) coordinate system or MGRS (Military Grid Reference System) grids are displayed. Each UTM grid is a square of 10 km per side.

**Table 1 plants-13-00970-t001:** Number of species present in the sampled 1000 m^2^ quadrat: total species (nW), gypsovag species (nGv) and gypsophiles + gypsoclines species (nGyGc). S1 = outcrops > 10 km^2^; S2 = outcrops 1–10 km^2^; S3 = outcrops < 1 km^2^. n = number of sampled quadrats; μ = mean species in the samples; std = standard deviation; s.d. = significant differences between group size.

		nW			nGv			nGyGc		
Size	n	μ	std	s.d.	μ	std	s.d.	μ	std	s.d.
**S1**	5	65.60	7.89	S2, S3	51.80	7.46	S2, S3	13.80	0.84	S2, S3
**S2**	15	52.47	7.76	S1, S3	41.07	6.99	S1, S3	10.93	2.46	S1, S3
**S3**	10	38.00	7.30	S1, S2	29.10	8.80	S1, S2	8.80	2.20	S1, S2

**Table 2 plants-13-00970-t002:** Comparison between the slopes (*z*-values) obtained for the group of gypsophytes (Gy) and gypsophytes + gypsoclines (GyGc) with the rest of the taxonomic and functional groups considered. W = Wilcoxon rank–sum; *z* = median difference between the two groups; H-L = Hodges–Lehmann estimate; r_B_ = rank-biserial correlation; CI r_B_ = confidence intervals for r_B_. In all cases, n = 30. Rank–biserial (r_B_) “large” values in **bold**. See Materials and Methods section for the abbreviations used for group_2.

							95% CI r_B_ Correlation	
Group_1	Group_2	W	z	*p*	H-L	r_B_	Lower (r_B_)	Upper (r_B_)
Gy_z	Poa_z	62.000	−3.507	<0.001	−0.067	**−0.733**	−0.873	−0.482
Gy_z	Ast_z	14.000	−4.494	<0.001	−0.118	**−0.940**	−0.973	−0.868
Gy_z	Ste_z	18.000	−4.314	<0.001	−0.117	**−0.917**	−0.963	−0.819
Gy_z	Ros_z	44.000	−3.877	<0.001	−0.065	**−0.811**	−0.912	−0.617
Gy_z	Gv_z	18.000	−4.412	<0.001	−0.113	**−0.923**	−0.965	−0.833
Gy_z	GvP_z	3.000	−4.720	<0.001	−0.141	**−0.987**	−0.994	−0.971
Gy_z	P_nGR_z	5.000	−4.679	<0.001	−0.092	**−0.978**	−0.990	−0.952
Gy_z	W_z	20.000	−4.371	<0.001	−0.069	**−0.914**	−0.961	−0.815
GyGc_z	Poa_z	57.000	−3.610	<0.001	−0.075	**−0.755**	−0.884	−0.518
GyGc_z	Ast_z	10.000	−4.576	<0.001	−0.122	**−0.957**	−0.981	−0.905
GyGc_z	Ste_z	20.000	−4.271	<0.001	−0.123	**−0.908**	−0.959	−0.800
GyGc_z	Ros_z	26.000	−4.247	<0.001	−0.072	**−0.888**	−0.949	−0.763
GyGc_z	Gv_z	19.000	−4.391	<0.001	−0.119	**−0.918**	−0.963	−0.824
GyGc_z	GvP_z	2.000	−4.741	<0.001	−0.148	**−0.991**	−0.996	−0.981
GyGc_z	P_nGR_z	2.000	−4.741	<0.001	−0.099	**−0.991**	−0.996	−0.981
GyGc_z	W_z	19.000	−4.391	<0.001	−0.076	**−0.918**	−0.963	−0.824

**Table 3 plants-13-00970-t003:** Comparison between the slopes (z-values) obtained for the group of gypsophytes (Gy) and gypsophytes + gypsoclines (GyGc) with the 10 randomly generated groups (POOL). W = Wilcoxon rank–sum; *z* = median difference between the two groups; H-L = Hodges–Lehmann estimate; r_B_ = rank–biserial correlation; CI r_B_ = confidence intervals for r_B_. In all cases, n = 30. Rank–biserial (r_B_) “large” values in **bold**. In all cases, n = 30.

							95% CI for Rank–Biserial Correlation	
Group_1	Group_2	W	z	*p*	H-L	r_B_	Lower (r_B_)	Upper (r_B_)
Gy_z	POOL1_z	43.000	−3.898	<0.001	−0.060	**−0.815**	−0.914	−0.624
Gy_z	POOL2_z	20.000	−4.371	<0.001	−0.066	**−0.914**	−0.961	−0.815
Gy_z	POOL3_z	2.000	−4.741	<0.001	−0.080	**−0.991**	−0.996	−0.981
Gy_z	POOL4_z	7.000	−4.638	<0.001	−0.075	**−0.970**	−0.987	−0.933
Gy_z	POOL5_z	8.000	−4.618	<0.001	−0.064	**−0.966**	−0.985	−0.924
Gy_z	POOL6_z	60.000	−3.548	<0.001	−0.034	**−0.742**	−0.877	−0.497
Gy_z	POOL7_z	20.000	−4.371	<0.001	−0.048	**−0.914**	−0.961	−0.815
Gy_z	POOL8_z	85.000	−3.034	0.002	−0.047	**−0.634**	−0.821	−0.327
Gy_z	POOL9_z	87.000	−2.822	0.005	−0.034	**−0.600**	−0.804	−0.270
Gy_z	POOL10_z	9.000	−4.597	<0.001	−0.065	**−0.961**	−0.983	−0.914
GyGc_z	POOL1_z	33.000	−4.103	<0.001	−0.064	**−0.858**	−0.935	−0.704
GyGc_z	POOL2_z	12.000	−4.535	<0.001	−0.070	**−0.948**	−0.977	−0.887
GyGc_z	POOL3_z	0.000	−4.782	<0.001	−0.086	**−1.000**	−1.000	−1.000
GyGc_z	POOL4_z	8.000	−4.618	<0.001	−0.082	**−0.966**	−0.985	−0.924
GyGc_z	POOL5_z	4.000	−4.700	<0.001	−0.071	**−0.983**	−0.992	−0.961
GyGc_z	POOL6_z	22.000	−4.330	<0.001	−0.037	**−0.905**	−0.957	−0.797
GyGc_z	POOL7_z	9.000	−4.597	<0.001	−0.055	**−0.961**	−0.983	−0.914
GyGc_z	POOL8_z	48.000	−3.795	<0.001	−0.049	**−0.794**	−0.903	−0.586
GyGc_z	POOL9_z	64.000	−3.466	<0.001	−0.037	**−0.725**	−0.869	−0.468
GyGc_z	POOL10_z	7.000	−4.638	<0.001	−0.071	**−0.970**	−0.987	−0.933

**Table 4 plants-13-00970-t004:** Comparison for the *z*-values in wide (wG) and narrow gypsophytes (nG).

	wG	nG
n	30	30
mean	0.194	0.161
	mean difference	*p*
*t* test	0.032	0.075
Bayes factor	0.874	No evidence for either equal or unequal means

**Table 5 plants-13-00970-t005:** Characterizing the environmental conditions of the investigated outcrops. Data for threatened species were sourced from Hernández Bermejo et al. [76], Andalusian Catalog Law 8 [77]), Cabezudo et al. [78], National Red List [79], and Mota et al. [35]. The abbreviations employed for threatened flora are as follows: Ch = *Coris hispanica*, Eb = *Euzomodendron bourgaeanum*, Ga = *Guiraoa arvensis*, Ha = *Helianthemum alypoides*, Lf = *Lycocarpus fugax*, Lo = *Linaria oligantha*, Lt = *Limonium tabernense*, Nt = *Narcissus tortifolius*, Pd = *Pteranthus dichotomus*, Re = *Rosmarinus eriocalix*, Sp = *Salsola papillosa*, Sv = *Santolina viscosa*, Tc = *Teucrium charidemi*, Tt = *Teucrium turredanum*. Abbreviations for the sites: Sorbas (SO), Venta de los Yesos (VY), Cueva de los Úbedas (CU), Alfaro (AL), Yesón Alto (YA), and Polopos (PO).

Site	SO	VY	CU	AL	YA	PO
Outcrop surface area (km^2^)	18.65	2.50	1.02	0.11	1.80	0.34
Size	Large (S1)	Medium (S2)	Medium (S2)	Small (S3)	Medium (S2)	Small (S3)
Geology (type of gypsum)	Upper Messinian (selenite)	Upper Messinian (selenite)	Upper Messinian (selenite)	Upper Messinian (gypsarenite)	Upper Messinian (selenite)	Upper Messinian (selenite)
Minimum altitude (m)	240	495	187	186	390	220
Maximum altitude (m)	660	626	256	265	661	336
Mean altitude (m)	450	561	222	226	526	278
Average annual precipitation (mm)	262	261	234	240	259	248
Distance to nearest outcrop (km)	4.41	12.03	14.32	6.79	6.79	4.41
Distance to source outcrop (km)	-	-	17.95	17.53	16.41	4.41
Average distance to the rest of the outcrops (km)	21.23	16.39	20.48	21.26	22.88	24.12
Percentage included in protected area	96.8	0	61.9	100	100	100
Percentage of the surface affected by mining concessions	95.1	77.8	100	0	100	0
Active quarries	Yes	Yes	Yes	No	No	No
Presence of abandoned quarries in the outcrop	Yes	Yes	Yes	No	No	Yes
Number of species present	350	200	148	109	132	125
Number of gypsophile species + (gypsoclines)	14 + (5)	13 + (4)	9 + (4)	4 + (3)	8 + (5)	10 + (3)
Number of gypsophile species in the set of samples + (gypsoclines)	13 + (5)	13 + (4)	9 + (4)	4 + (3)	8 + (5)	10 + (3)
Maximum number of gypsophile species in the samples + (gypsoclines)	12 + (3)	11 + (4)	8 + (4)	4 + (3)	8 + (5)	9 + (3)
Threatened species	Ch, Ga, Ha, Lo, Nt, Sv, Tt	Ch, Lo, Sv, Re	Ch, Sp, Sv	Ch, Eb, Lt, Sp, Pd	Ch, Lf, Lo, Sp, Sv	Ch, Ha, Nt, Re, Sv, Tc, Tt

**Table 6 plants-13-00970-t006:** Species’ assemblages considered for SAR (species–area relationship) analyses.

**Gy**	Gypsophytes (species that exclusively grow on gypsum)
**GyGc**	Gypsophytes and gysopclines (species with a strong preference for gypsum, but not exclusive)
**Poa**	*Poaceae* family representatives
**Ast**	*Asteraceae* family representatives
**Ste**	*Stellarietea mediae* class representatives
**Ros**	*Rosmarinetea officinalis* class representatives
**Gv**	Gypsovags, i.e., species that can grow on different types of soils
**GvP**	Perennial gypsovags
**P_nGR**	Non-gypsophile species within the *Rosmarinetea officinalis* class
**W**	All species recorded in the sample plots
**wG**	Gypsophytes widely distributed, found throughout Spain and beyond
**nG**	Locally or regionally endemic gypsophytes

## Data Availability

Any data that is required will be provided.

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
