# Peer review of "Shipwrecked on the Rock, or Not Quite: Gypsophytes and Edaphic Islands"

_plants, 2024, doi:10.3390/plants13070970_

Round 1

Reviewer 1 Report

Comments and Suggestions for Authors

Review of " Shipwrecked on the rock, or not quite: gypsophytes and edaphic islands " submited to Plants.

This paper addresses the role of gypsum outcrops in SE Spain as edaphic islands and the role of outcrop size on plant richness. I think that the subject of this paper deserves attention and may result interesting for the readers of Plants.

However, I have several concerns with the manuscript in its present form.

First, I miss a set of clear and explicit hypotheses, and a clear description of the statistical tools that would be employed to test them. These specific hypotheses should have been explained and/or justified in the introduction. At least, these specific hypotheses should have been framed within the general models considered for island biogeography (see, e.g. Whittaker and Fernández-Palacios 2007, section 4.2) or for edaphic islands in particular (e.g., Méndez-Castro et al. 2021). In fact, most of the Introduction is quite general and there is scarce material about the problems to characterize edaphic islands. In addition, the selection of species assemblages (Table 6) should be justified in the Introduction.

Second, I think that the statistical approach could be severely improved by considering models which include covariates other than outcrop area to account for the variation of species richness. In fact, most of these covariates have been computed by the authors (see Table 5 in the manuscript) but are not consistently employed in the statistical analyses. See Méndez-Castro et al. 2021 for a possible (not unique) example of a more appropriate approach.

Finally, the writing of the manuscript should be profoundly revised, avoiding convoluted wording which generates ambiguity and/or confusion in many places (see minor comments below) and being more precise to avoid ambiguity (e.g., in L345-346: " only the plants included within the sampling plots were considered, **either entirely in some cases or partially** " or L366: " they were included **in some analyses**").

I hope that this review helps the authors to improve their paper. Other comments follow.

Minor comments

Introduction.

L43: z is the slope of the (log) linearized model, i.e., log(S) = log(c) + z log(A)

L52: You have only explained Arrhenius model. You should explain Gleason model.

L53: Do you mean "c" or "log (c)" here?

L58: Define what do you mean by "insularity" and "degree of insularity". In fact, [16]Scheiner (2003) does not mention neither of those concepts. Do you mean "isolation"? (beware that Sheiner does not mention isolation neither).

L59-60: "In general, higher values of z *are interpreted* as indicating higher isolation, "

L59-61: Beware: isolation is not necessarily related to island area. This sentence is incorrect.

76-77: Many other authors before Matthews et al. 2016 [25] have shown that ISAR and SAC are not equivalent. For example, Sheiner 2003. In any case, this lack of equivalence is not related to the island being edaphic or other type's.

L82-86 (from "Kruckeberg.."): Too vague and unrelated to the previous paragraph. Remove.

L90-94: Please, specify which are the "first" and "second" hypotheses here.

L93-94: I cannot see the logic of this argument. Wouldn't it be the other way round, i.e. that the insular effect of gypsum outcrops would be *more* pronounced for gypsophytes than for generalist species?

L96-97: Please, clarify this. Which "authors' hypothesis" do you refer to? By the way, an observation may support a hypothesis, but not the other way round. Do you mean that your hypothesis follow or come from the observation by Izco? In general, a concise enumeration of individual hypotheses would improve the assessment of this paper.

L99: What are the "normal populations" for gypsohphytes? Do you mean their core populations?

L101: You have not described your approach.

L101-102. As you explained in L56-61, the interpretation of z as a measure of isolation is widely established in the scientific literature, so explain better what is innovative about this.

L103: What do you mean by "the differential insular nature" and how do you measure it? By the way, which ou "these outcrops"? You should have explained this in the introduction.

L105: It seems that ypu refer here to stenochory (please, define it) not to endemicity. Remove "endemicity".

L105-106: "wide-area" and "narrow-area gypsophytes".

Results

L114-117: There are not abbreviations for the localities in the plots.

L120: Table 1 shows differences among outcrop size groups, not differences among species groups according to their degree of gypsophily!!

L131: What are the values tabulated in the column "s.d."? If these are results from a post-hoc test, you should explain that "1" stands for "S1", etc and, in addition, you should describe this test in "methods".

L132: " yielded a mean value for the z exponent of... "

L133-137: These ar not "results" but "discussion". Move this sentence to that subsection.

L141-145: I miss here (and in Table2) a direct comparison of Gy vs Gy+Gc. In any case, as you are performing multiple tests (multiple comparisons) , you should account for this and present adjusted p-values (e.g., false discovery rate, Holm's, etc).

L150-153: Please, indicate where the reader could find the meaning of the groups abbreviations.

L162-163: You did not describe the multiple range test in "methods". You did not describe here the 9 homogeneous groups.

L186-187 (legend of Fig. 2): You should explain the different colors of the box plots.

L188: Which are "these same groups"? The wG and NG that you described in the previous sentence?

Discussion

L198: "This isolated result".

L199: Rewrite this. In fact, your models describe richness as a function of island size, so the outcome is as expected.

Methods

L288-289: You should give more details about these outcrops here. For example: are there any other outcrops in the area? Are there any gypsum soils among the depicted outcrops? Is there habitat heterogeneity within the outcrops?

L315-317: The term "target effect" is not mentioned in the main text. You should include it in Table 5 (and should use it in your models!).

L327: Indicate the number of subplots per plot, how where they placed and how they were employed to estimate species richness..

L334-338: Instead of referring all time to "power law regression", you should better explain the methods as the fitting of an ordinary least square regression, i.e., log(S) ~ log(c) + z log(A).

L340: If "the latter parameter" was the surface area, call this "the surface area". Please explain the method of Ochotorena et al.

L342-343: To ascertain if there were significant differences in species richness among what? If you want to test the effect of size area on richness, a t-test about z = 0 would be enough. You don't need to subjectively classify the outcrops in three (why not two or four?) categories.

L345-349:  This description of the analyses is too much confusing. You should have described before what type of species and groups of species, and based on what, you would be considering in your analyses.

L354-356: You are describing only two of three types (gypsophiles, gypsoclines, and gypsovags).

L357: "and wide-area gypsophytes".

L358: To compare what means?

L375-379: Although Lomolino (1989) cautions against ignoring differences of species pools when comparing SACs, I think it is more problematic comparing artificial species pools. In fact, you may include the size of the species pool as a covariate to account for this but you cannot control the effects of the subjective grouping of species in similar-sized species pools.

L386-391: You may discuss this in the "discussion", but not here in "methods".

L395: The comparison of z values, among which SAC curves? Please, detail. This is important to assess whether you need "simulations" or you may employ other inferential techniques.

L395-400: You should describe what null model did you simulate and explain how did you build your samples.

Comments on the Quality of English Language

The writing of the manuscript should be profoundly revised, avoiding convoluted wording which generates ambiguity and/or confusion in many places (see "minor comments" and being more precise to avoid ambiguity (e.g., in L345-346: " only the plants included within the sampling plots were considered, **either entirely in some cases or partially** " or L366: " they were included **in some analyses**").

Author Response

Reviewer 1.

Comments and Suggestions for Authors

Review of " Shipwrecked on the rock, or not quite: gypsophytes and edaphic islands " submited to Plants.

This paper addresses the role of gypsum outcrops in SE Spain as edaphic islands and the role of outcrop size on plant richness. I think that the subject of this paper deserves attention and may result interesting for the readers of Plants.

Response: Dear reviewer, we greatly appreciate the thorough review you have dedicated to our manuscript. We are honored that it has warranted your diligent effort to improve it. Since we have followed most of your suggestions, we believe this goal has been achieved, but we will address any other concerns that may remain. In the few cases where we have not implemented your suggestions, we have hereby justified our reservations. However, as you will see, this circumstance has been exceptional.

However, I have several concerns with the manuscript in its present form.

First, I miss a set of clear and explicit hypotheses, and a clear description of the statistical tools that would be employed to test them. These specific hypotheses should have been explained and/or justified in the introduction. At least, these specific hypotheses should have been framed within the general models considered for island biogeography (see, e.g. Whittaker and Fernández-Palacios 2007, section 4.2) or for edaphic islands in particular (e.g., Méndez-Castro et al. 2021). In fact, most of the Introduction is quite general and there is scarce material about the problems to characterize edaphic islands.

Response: We consider that with the changes made, the central hypotheses of the manuscript are now clearly expressed. The references mentioned for the conceptual framework are included in the introduction, and we consider them key to understanding the purpose of this manuscript. The introduction is a crucial section in any scientific article, and in it, we have attempted to link SAC, SAR, and ISAR, all of which are key aspects. Also, islands and so-called continental islands. The sequence of ideas associated with this development is undoubtedly complex and likely depends heavily on the objectives being pursued. Therefore, we agree with your observations and have tried to adapt to them regarding the plants linked to gypsum soils.

In addition, the selection of species assemblages (Table 6) should be justified in the Introduction.

Response: Done.

Second, I think that the statistical approach could be severely improved by considering models which include covariates other than outcrop area to account for the variation of species richness. In fact, most of these covariates have been computed by the authors (see Table 5 in the manuscript) but are not consistently employed in the statistical analyses.

Response: Truly, this is perhaps one of the comments that has surprised us the most. There is no doubt that there are many factors that can be used as covariates in relation to species’ richness in our plots and to understand the values of z. That can be a very interesting idea to expand this research because, as you rightly point out, many of these covariates are included in the table you mention. However, we would like to point out that according to the approach of our research and the hypotheses proposed, it does not make sense to use covariates. You should consider that all comparisons made are pairwise, so statistically in this case covariates are nullified, since they are similar for each of the compared samples. However, we will explore the research approach so appealingly suggested by you

See Méndez-Castro et al. 2021 for a possible (not unique) example of a more appropriate approach.

Response: We also consider the research by Méndez-Castro et al to be very valuable, although with very different objectives compared to ours. They have in common that they deal with edaphic islands, but from there everything seems different. The fact that you consider it more appropriate than what we offer is something we must respect, considering that in our sincere opinion, they are not comparable. This comment, along with what was expressed a little earlier, encourages us, however, to try to expand our approach inspired by that and other equally valuable observations that the referees have made. Thank you again for your suggestions.

 Finally, the writing of the manuscript should be profoundly revised, avoiding convoluted wording which generates ambiguity and/or confusion in many places (see minor comments below) and being more precise to avoid ambiguity (e.g., in L345-346: " only the plants included within the sampling plots were considered, **either entirely in some cases or partially** " or L366: " they were included **in some analyses**").

Response: Refer to the end of this document for our latest comment on this matter.

I hope that this review helps the authors to improve their paper. Other comments follow.

Minor comments

Introduction.

L43: z is the slope of the (log) linearized model, i.e., log(S) = log(c) + z log(A)

Response: Fixed.

L52: You have only explained Arrhenius model. You should explain Gleason model.

Response: We greatly appreciate the suggestion, but since this model is not utilized throughout the research, it seems unnecessary to divert the reader's attention towards it.

L53: Do you mean "c" or "log (c)" here?

Response: We refer to c, which in many of the cited references is interpreted as a surrogate for species richness.

L58: Define what do you mean by "insularity" and "degree of insularity". In fact, [16]Scheiner (2003) does not mention neither of those concepts. Do you mean "isolation"? (beware that Sheiner does not mention isolation neither).

Response: That's also a good observation. Few authors define it when addressing SAR, but MENDEZ-CASTRO ET AL. do. So, we have placed this reference, of great value to our research, in the appropriate place in the text. Thank you.

L59-60: "In general, higher values of z *are interpreted* as indicating higher isolation, "

Response: Again, another very appropriate comment that we have taken into account. Fixed.

L59-61: Beware: isolation is not necessarily related to island area. This sentence is incorrect.

Response: We almost entirely agree with this observation, although there are authors who have spoken of the small-island area effect. However, we have not corrected the sentence since we do not see that it establishes the link between area and isolation. Yet, we are open to any comments that could clarify this aspect.

76-77: Many other authors before Matthews et al. 2016 [25] have shown that ISAR and SAC are not equivalent. For example, Scheiner 2003. In any case, this lack of equivalence is not related to the island being edaphic or other type's.

Response: We add Scheiner (2003) to the references here. We find it extremely important to acknowledge the contributions of our colleagues, especially if they are pioneering.

L82-86 (from "Kruckeberg.."): Too vague and unrelated to the previous paragraph. Remove.

Response: We don't see it the same way, We´re sorry. The idea here is to highlight which studies dealt early on with continental islands, especially in the case of plants.

L90-94: Please, specify which are the "first" and "second" hypotheses here.

Response: The text has been rewritten to enhance comprehension and to make the hypotheses much clearer.

L93-94: I cannot see the logic of this argument. Wouldn't it be the other way round, i.e. that the insular effect of gypsum outcrops would be *more* pronounced for gypsophytes than for generalist species?

Response: We perfectly understand the interpretation you suggest. Perhaps, when it comes to logic, this argument may seem paradoxical, but the sentence precisely reflects what is meant. And that is precisely what Izco reflects in his comment (see below).

L96-97: Please, clarify this. Which "authors' hypothesis" do you refer to? By the way, an observation may support a hypothesis, but not the other way round. Do you mean that your hypothesis follow or come from the observation by Izco? In general, a concise enumeration of individual hypotheses would improve the assessment of this paper.

Response: Indeed, our hypothesis, as previously discussed, aligns with Izco's observation. Furthermore, we have striven to make everything clearer now.

L99: What are the "normal populations" for gypsohphytes? Do you mean their core populations?

Response: Yes, "core" is a more suitable term. We completely agree.

L101: You have not described your approach.

Response: We have reformulated the sentence.

L101-102. As you explained in L56-61, the interpretation of z as a measure of isolation is widely established in the scientific literature, so explain better what is innovative about this.

Response: So far, no one had bothered to apply these ideas to gypsum outcrops, although there are many scientific articles that take for granted that gypsum share some characteristics of island systems.

L103: What do you mean by "the differential insular nature" and how do you measure it? By the way, which ou "these outcrops"? You should have explained this in the introduction.

Response: That the values of z are different for gypsophytes vs gypsovags and that this parameter is precisely an indicator of this. We believe it has been clarified now; also, in the 'Introduction'.

L105: It seems that ypu refer here to stenochory (please, define it) not to endemicity. Remove "endemicity".

Response: Indeed, stenochory and endemism are not exactly the same, but the concept of endemism makes no sense if it does not refer to a distribution area. We see no problem in keeping both concepts linked. Endemism is a manifestation of stenochory, although it has also been used to refer to the restriction of some species to special soils, thus linking the concept with ecological aspects. Undoubtedly, an interesting topic for discussion, but the origins of the concept primarily link it to the species' area.

L105-106: "wide-area" and "narrow-area gypsophytes".

Response: That's correct, but if you read references about gypsum plants, you will find that it's common to refer in this abbreviated way to the idea that the reviewer has expressed by adding the word "area". Anyway, we have fixed it.

Results

L114-117: There are not abbreviations for the localities in the plots.

Response: Fixed. Thanks

L120: Table 1 shows differences among outcrop size groups, not differences among species groups according to their degree of gypsophily!!

Response: Fixed. Thanks again.

L131: What are the values tabulated in the column "s.d."? If these are results from a post-hoc test, you should explain that "1" stands for "S1", etc and, in addition, you should describe this test in "methods".

Response: As you will see in the manuscript, we have made significant changes to the statistical data processing and have also modified the corresponding text sections. Although the results have hardly been affected, I believe your feedback has substantially improved this important aspect of the manuscript.

L132: " yielded a mean value for the z exponent of... "

Response: Added.

L133-137: These ar not "results" but "discussion". Move this sentence to that subsection.

Response: Done.

L141-145: I miss here (and in Table2) a direct comparison of Gy vs Gy+Gc. In any case, as you are performing multiple tests (multiple comparisons) , you should account for this and present adjusted p-values (e.g., false discovery rate, Holm's, etc).

Response: Direct comparisons between Gy and Gy+Gc have not been made because we understand that they are not of interest in the research and may distract the reader's attention. The Cy+Gc group has been created because the number of species it includes is slightly higher, and as stated throughout the text, it is important for the values of n to be approximately equal in the comparisons. There are no significant differences between the two groups, as expected. This is also evident when paying attention to Figure 2, which has been substantially improved now. Regarding the use of post-hoc tests, please note that key aspects of the statistical analysis have been modified. Considering that the comparisons for z-values are paired, the rank-biserial correlation (rB) should now be taken into account.

L150-153: Please, indicate where the reader could find the meaning of the groups abbreviations.

Response: Done.

L162-163: You did not describe the multiple range test in "methods". You did not describe here the 9 homogeneous groups.

Response: Please refer to previous comments regarding this matter.

L186-187 (legend of Fig. 2): You should explain the different colors of the box plots.

Response: This figure has been replaced by a much clearer one.

L188: Which are "these same groups"? The wG and NG that you described in the previous sentence?

Response: The reference to Table 2 helps clarify this point.

Discussion

L198: "This isolated result".

Response: Fixed.

L199: Rewrite this. In fact, your models describe richness as a function of island size, so the outcome is as expected.

Response: We have rewritten the paragraph. Thank you.

Methods

L288-289: You should give more details about these outcrops here. For example: are there any other outcrops in the area? Are there any gypsum soils among the depicted outcrops? Is there habitat heterogeneity within the outcrops?

Response: Figure 3 is quite explicit regarding the first two questions. Also, the reference to Mota et al. 2011, which is very useful for addressing the third question along with the data collected in Table 5. There are more outcrops in the province of Almería, in the southeastern part of Spain, and in Spain overall, but not in the study area.

L315-317: The term "target effect" is not mentioned in the main text. You should include it in Table 5 (and should use it in your models!).

Response: Certainly, it is indeed a very interesting concept upon which the studied outcrops could serve as a basis for testing its usefulness. We are interested in this idea, but we believe that its implementation would not add much more information to the research as it is currently framed. Therefore, we have chosen to exclude mentions to it in the manuscript.

L327: Indicate the number of subplots per plot, how where they placed and how they were employed to estimate species richness.

Response: All that information is exhaustively detailed, in writing and with an illustrative scheme, in the provided reference. This reference is open access, so any reader can conveniently consult it.

L334-338: Instead of referring all time to "power law regression", you should better explain the methods as the fitting of an ordinary least square regression, i.e., log(S) ~ log(c) + z log(A).

Response: This seems to be primarily a matter of personal preference, doesn’t it? Perhaps the suggested expression may be more purist, but in this case, we don't find an irresistible argument.

L340: If "the latter parameter" was the surface area, call this "the surface area". Please explain the method of Ochotorena et al.

Response: This seems like a very similar case, given the repetitiveness of the expression, to the issue discussed in the preceding point. Nonetheless, we are happy to make the change. We have also outlined the basic aspects of the Ochoterena et al. method.

L342-343: To ascertain if there were significant differences in species richness among what? If you want to test the effect of size area on richness, a t-test about z = 0 would be enough. You don't need to subjectively classify the outcrops in three (why not two or four?) categories.

Response: The differences established between the size of the outcrops are based on orders of magnitude, so they are neither two nor four. The proposed approach is indeed possible, but it adds more detail to the analyses.

L345-349:  This description of the analyses is too much confusing. You should have described before what type of species and groups of species, and based on what, you would be considering in your analyses.

Response: Yes, it's confusing. We believe we've fixed it by rephrasing the paragraph in question. Thank you.

L354-356: You are describing only two of three types (gypsophiles, gypsoclines, and gypsovags).

Response: We regret to express our disagreement here, but in this case, it is not accurate to say that only two of the three functional types of plants have been described based on their link to gypsum soils. Gypsophytes, gypsoclines, and gypsovags are described in the paragraph.

L357: "and wide-area gypsophytes".

Response: Done.

L358: To compare what means?

Response: Clarified.

L375-379: Although Lomolino (1989) cautions against ignoring differences of species pools when comparing SACs, I think it is more problematic comparing artificial species pools. In fact, you may include the size of the species pool as a covariate to account for this but you cannot control the effects of the subjective grouping of species in similar-sized species pools.

Response: The grouping was not subjective; it was random.

L386-391: You may discuss this in the "discussion", but not here in "methods".

Response: We do not consider this paragraph to be part of the discussion. Here, we simply acknowledge a benchmark or threshold value for z as 0.17, which was established by other researchers, as recognized in the text.

L395: The comparison of z values, among which SAC curves? Please, detail. This is important to assess whether you need "simulations" or you may employ other inferential techniques.

Response: Those reflected in Tables 2 and 3. This is now indicated in the text, although it seemed obvious to us. We consider now the statistical inference to be robust enough to support the conclusions. Very related analyses, among them are those from the previous version of the manuscript that you are familiar with, using other methods (e.g., Bayesian statistics), yield the same results. As previously mentioned, and despite that, we have changed the statistical approach of the manuscript, always with the premise of improving it so that the results obtained are rigorous. We hope you will appreciate this effort

L395-400: You should describe what null model did you simulate and explain how did you build your samples.

Response: See previous comments.

Comments on the Quality of English Language

The writing of the manuscript should be profoundly revised, avoiding convoluted wording which generates ambiguity and/or confusion in many places (see "minor comments" and being more precise to avoid ambiguity (e.g., in L345-346: " only the plants included within the sampling plots were considered, **either entirely in some cases or partially** " or L366: " they were included **in some analyses**").

Response: This is a commonplace place visited countless times by Spanish scientists. We cannot help but be struck by this type of criticism, almost always unjust, not because of our level of English, but because they assume that we have not cared about language. We are aware that great scientists in history, like Darwin himself, were great writers. For this reason, we believe that clarity in the exposition of arguments is extremely valuable and, for this reason, we always ask that our language be reviewed by very capable people. In this case, our text has been meticulously reviewed by a person whose native language is English. It is striking that it is assumed without evidence that being Spanish means we do not care about language or that surnames that do not sound British or American imply that someone is not native. Sometimes we have the feeling that, although few, scientists whose native language is English are not aware of the enormous advantage they enjoy. We confess, we are very envious of that condition. But, also, as Spanish speakers, we are aware of the differences between Spanish in America and other parts of the world. As we have traveled a little, we have the feeling that English also suffers the effects of geography. In any case, once again we show our gratitude for your generous help with language that we will try to improve.

Reviewer 2 Report

Comments and Suggestions for Authors

The title of this paper took my attention at first glance. The study fits to the basic ecological theories. The limited distribution of these deposits and their interesting ecology, as well as their conservation significance as a habitat included in the Habitats Directive are the reason why such studies are of great importance. It’s obvious from the references that the authors have long practice related to the gypsum deposits and their vegetation. I have no doubt that this article deserves publishing, but by my opinion it needs some edits to make it better and easier for readers’ understanding. Below, I offer several suggestions for this.

It would be good to emphasize what is the main message of the study: to confirm the theory of Species-Area Relationships or to describe the role of gypsum islands as a filter for different groups of species?

Key words include “successional chronosequence” which has been a subject of another study of the authors but not a matter of the current one. I suggest to substitute it with gypsovags.

The introductory part will benefit from a better interpretation of the difference between Species Accumulation Curves and Species-Area Relationships. I do not understand why the insular effect for gypsophytes is less pronounced than for generalists (line 93)? And how this has to be interpret with the dispersal capacity. Finally, instead of research questions the authors assess their research approach as innovative but actually they apply old methods to obtain new information about gypsum deposits in the country.

Line 205: instead of ‘differentially permeable’ it’s perhaps better to mention ‘differentially capable to host’

Line 210: “the mentioned matrix is permeable to a high percentage of species” – surrounding matrix or gypsum islands?

Lines 214 - 215: some (not all) gypsopile species are also absent here, such as Lepidium subulatum and Frankenia thymifolia (should be in italic). May be this finding is related to some disturbances not to the propagule source? Lower z-values for gypsophile and gypsocline species might be a result of their low number in the species pool?

Lines 292-294: Instead of explanation what “c” is, I would recommend to explain how it was calculated.

Line 330: ‘The presence of various vascular plant species’ or ‘The presence of all vascular plant species’? The next sentence (lines 331-332) is a part of the results.

Lines 339-349: It is not clear why the regression model uses number of species obtained in other study. Estimation of the surface area could be briefly mentioned to help readers not to search in additional literature. It would be good to add the dates of the field survey in the methods. My greatest concern is about the species grouping considered for SAR. As a mixture between functional groups, taxonomic groups and syntaxonomic indicators, some species could overlap and bias the results. Same is with the gypsovags and non-gypsophile species within the Rosmarinetea officinalis.

Line 402: Please, explain what are “experimental and control groups” and “before and after the intervention”. Further in this paragraph, the meaning of Cohen d statistic related to the ‘effect’ is explained. Probably ‘effect’ was used instead of ‘difference between groups’.

Results section actually includes mixture of results and discussion (e.g. lines 133-147).

In the Conclusion section the authors indicate rather expected and trivial statement that gypsophiles are better adapted to live on gypsum islands, while gypsovags have no specific adaptations to this environment. It would be interesting to indicate whether there are gypsophiles + gypsoclines in the surrounding territories of gypsum islands, which presence will affect the species propagules pool. Between lines 434-443 there are comments on the species peculiarities which are not a matter of this study, however the paragraph will benefit if here authors add at lest one or two examples to support the comments.

Line 463: Monitoring and Applied Scientific Research

Lines 625-628: Please, remove hyphenation (flo-ra, multi–scale)

Tables 2 and 3: Please, explain what means n=30.

Table 5: What means distance to source outcrop? How it is possible 100% of protected area to be included 100% in mining concession? I did not found the term "target effect" [27], in the main text (see caption of the table).  

Figure 1 lacks the labels of the sites and the measure for the area (on the abscissa).

Author Response

Reviewer 2.

Comments and Suggestions for Authors

The title of this paper took my attention at first glance. The study fits to the basic ecological theories. The limited distribution of these deposits and their interesting ecology, as well as their conservation significance as a habitat included in the Habitats Directive are the reason why such studies are of great importance. It’s obvious from the references that the authors have long practice related to the gypsum deposits and their vegetation. I have no doubt that this article deserves publishing, but by my opinion it needs some edits to make it better and easier for readers’ understanding. Below, I offer several suggestions for this.

Response: Dear reviewer, thank you very much for the attention you have paid to our manuscript. We are confident that your insightful comments have greatly contributed to its improvement. Virtually all of them have been addressed, and we hope that our responses will help clarify any queries and questions that the manuscript may have raised.

It would be good to emphasize what is the main message of the study: to confirm the theory of Species-Area Relationships or to describe the role of gypsum islands as a filter for different groups of species?

Response: We like the idea. We have tried to do it this way with the corrections we have incorporated into the text.

Key words include “successional chronosequence” which has been a subject of another study of the authors but not a matter of the current one. I suggest to substitute it with gypsovags.

Response: Done

The introductory part will benefit from a better interpretation of the difference between Species Accumulation Curves and Species-Area Relationships. I do not understand why the insular effect for gypsophytes is less pronounced than for generalists (line 93)?

Response: At this point in the manuscript, this is just a hypothesis that is tested later on. In any case, we have tried to reflect more clearly the hypotheses that are proposed.

And how this has to be interpret with the dispersal capacity.

Response: It is evident that dispersal capacity is an important aspect in relation to species distribution. In the case of gypsophile plants, all references we have found indicate that, being endemics, the dispersal capacity of these plants is limited. As far as we know, there is no direct evidence of this. And that happens even though we have read everything that has been published about this type of flora.  In those thorough readings, we have come across very marginal information indicating that partridges eat seeds of some of them. This is mentioned in the text and simply highlights that this issue has been addressed too casually. In this case, our message is: let's be cautious with such assertive claims without providing any evidence. Furthermore, following this comment, we have sought additional information on this important yet under-researched but crucial aspect. We have incorporated this new information into the manuscript. However, this research does not aim, not even secondarily, to prove anything about the dispersal capacity of gypsophile plants. It simply points out that generalizing their limited dispersal capacity without empirical support is unreasonable and that there is evidence, albeit not very robust, that invites further study and deeper reflection.

Finally, instead of research questions the authors assess their research approach as innovative but actually they apply old methods to obtain new information about gypsum deposits in the country.

Response: This is a rather unfair assertion regarding the research on species-area relationships and the biogeography of islands or insular systems. 'Old'? Is the theory of evolution old? Is heliocentrism old? Do theories always worsen as they age? Dear reviewer, please conduct a bibliographic search on these and other topics on which this research is based, and you will see that it is a fertile field of research even today. Let us repeat here a phrase included in the manuscript with which we completely agree about SAR: 'is one of the few principles in ecology'. Nevertheless, we must admit that we must have done something wrong to lead you to this conclusion. In this way, you will see that we have modified the introduction trying to clarify which aspects we consider innovative in our research. However, if you know of any reference that may have escaped our attention, in which an attempt has been made to demonstrate, with some kind of evidence, that gypsum outcrops act as insular systems for plants, please let us know urgently. That reference will be very welcome if it goes beyond asserting this insular character and presents it as absolutely proven.

Line 205: instead of ‘differentially permeable’ it’s perhaps better to mention ‘differentially capable to host’.

Response: You will notice multiple changes in the manuscript following this recommendation.

Line 210: “the mentioned matrix is permeable to a high percentage of species” – surrounding matrix or gypsum islands?

Response: It refers to the surrounding matrix, yes. We consider that this is clear in the text in its current form.

Lines 214 - 215: some (not all) gypsopile species are also absent here, such as Lepidium subulatum and Frankenia thymifolia (should be in italic). May be this finding is related to some disturbances not to the propagule source? Lower z-values for gypsophile and gypsocline species might be a result of their low number in the species pool?

Response: This is a good observation, which is why we have taken great care to make comparisons with ecological and functional groups of species with a number of components or members (pool) quite similar to that of gypsophile species. It is a fundamental aspect of the manuscript.

Lines 292-294: Instead of explanation what “c” is, I would recommend to explain how it was calculated.

Response: See line 43 in the original manuscript.

Line 330: ‘The presence of various vascular plant species’ or ‘The presence of all vascular plant species’? The next sentence (lines 331-332) is a part of the results.

Response: Indeed, it concerns all vascular species. Corrected.

Lines 339-349: It is not clear why the regression model uses number of species obtained in other study.

Response: When this information is available, it is better to mention it. This makes floristic catalogs available to other researchers and acknowledges the work of other colleagues. Obviously, during the many years of study we have dedicated to gypsum, we have confirmed most of the species listed in those catalogs, but it does not seem relevant to this research. In fact, that information is reflected and expanded upon in the doctoral thesis of one of the authors of this manuscript.

Estimation of the surface area could be briefly mentioned to help readers not to search in additional literature.

Response: We have added some details about the procedure, but we do not wish to unnecessarily lengthen the manuscript.

It would be good to add the dates of the field survey in the methods. Done. My greatest concern is about the species grouping considered for SAR. As a mixture between functional groups, taxonomic groups and syntaxonomic indicators, some species could overlap and bias the results. Same is with the gypsovags and non-gypsophile species within the Rosmarinetea officinalis.

Response: Here, a random sampling strategy with replacement was applied, something common in the study of finite sample populations that, in no way, affects the results.

Line 402: Please, explain what are “experimental and control groups” and “before and after the intervention”. Further in this paragraph, the meaning of Cohen d statistic related to the ‘effect’ is explained. Probably ‘effect’ was used instead of ‘difference between groups’.

Response: Done.

Results section actually includes mixture of results and discussion (e.g. lines 133-147).

Response: Fixed. We have moved this paragraph to the "Discussion" section.

In the Conclusion section the authors indicate rather expected and trivial statement that gypsophiles are better adapted to live on gypsum islands, while gypsovags have no specific adaptations to this environment. It would be interesting to indicate whether there are gypsophiles + gypsoclines in the surrounding territories of gypsum islands, which presence will affect the species propagules pool. Between lines 434-443 there are comments on the species peculiarities which are not a matter of this study, however the paragraph will benefit if here authors add at lest one or two examples to support the comments.

Line 463: Monitoring and Applied Scientific Research MONITORING AND EVALUATION OF ENVIRONMENTAL RESTORATION OF THE MINING CONCESSIONS OF LOS YESARES, ANA MARÍA MORALES, AND EL CIGARRÓN

Lines 625-628: Please, remove hyphenation (flo-ra, multi–scale)

Response: Corrected for flora, but the forms 'multi-scale' and 'multiscale' are widely accepted and understandable in English.

Tables 2 and 3: Please, explain what means n=30.

Response: In statistics, "n" typically refers to the sample size, which is the number of observations or elements in a dataset.

Table 5: What means distance to source outcrop?

Response: In this case, the larger outcrop is considered as the source, which shares the same gypsophile species with those related to it, traditionally interpreted as belonging to the same biogeographic unit or province.

How it is possible 100% of protected area to be included 100% in mining concession?

Response: “Reality may be stranger than fiction”. That's how it is. And just as incongruent are the nature protection figures in Spain.

I did not found the term "target effect" [27], in the main text (see caption of the table).

Response: Fixed.

Figure 1 lacks the labels of the sites and the measure for the area (on the abscissa).

Response: Fixed.

Reviewer 3 Report

Comments and Suggestions for Authors

I read this paper with high interest, is very interesting and well written, I suggest only to authors to consult and possible cite the following papers:

Landi, M., Frignani, F., Lazzeri, C., & Angiolini, C. (2009). Abundance of orchids on calcareous grasslands in relation to community species, environmental, and vegetational conditions. Russian Journal of Ecology, 40, 486-494. Chiarucci, A. (2003). Vegetation ecology and conservation on Tuscan ultramafic soils. The Botanical Review, 69(3), 252-268. D’Antraccoli, M., Roma-Marzio, F., Carta, A., Landi, S., Bedini, G., Chiarucci, A., & Peruzzi, L. (2019). Drivers of floristic richness in the Mediterranean: a case study from Tuscany. Biodiversity and Conservation, 28(6), 1411-1429.

Author Response

Reviewer 3.

Comments and Suggestions for Authors

I read this paper with high interest, is very interesting and well written, I suggest only to authors to consult and possible cite the following papers:

Landi, M., Frignani, F., Lazzeri, C., & Angiolini, C. (2009). Abundance of orchids on calcareous grasslands in relation to community species, environmental, and vegetational conditions. Russian Journal of Ecology40, 486-494. Chiarucci, A. (2003). Vegetation ecology and conservation on Tuscan ultramafic soils. The Botanical Review69(3), 252-268. D’Antraccoli, M., Roma-Marzio, F., Carta, A., Landi, S., Bedini, G., Chiarucci, A., & Peruzzi, L. (2019). Drivers of floristic richness in the Mediterranean: a case study from Tuscany. Biodiversity and Conservation28(6), 1411-1429.

Response: Thank you for the excellent evaluation you have made of our manuscript. Two of the references you mention seem interesting to us and have been included. Although the article by D’Antraccoli et al. (2019) is of great interest and a valuable contribution, as it does not explicitly address insular systems or soils with special characteristics, we found it difficult to include it in the text of the article. Nevertheless, we have included the other two references as well as another highly valuable reference authored by some of the researchers suggested to us. Specifically:  Chiarucci, A., Bacaro, G., Triantis, K. A., & Fernandez-Palacios, J. M. (2011). Biogeographical determinants of pteridophytes and spermatophytes on oceanic archipelagos. Systematics and Biodiversity, 9(3), 191-201.

Reviewer 4 Report

Comments and Suggestions for Authors

In general, the manuscript is well organized and written, although at times it is somewhat convoluted and unclear. The research addresses a topic of interest but is repetitive. In this case the authors use Species-Area Relationships (SAR) to analyze the performance of gypsum outcrops as edaphic islands, differentiating between different degrees of gypsophily. This methodology is not new but has a long tradition in Population and Community Ecology. In this sense, the authors use the most popular postulates from scientific literature. This work can be enriched if more attention is paid to the parameter c, and also the heterogeneity of the habitat, the susceptibility of species to extinction or the effect of the size of the plots on sampling is taken into account. Ultimately, it cannot be ruled out that the origin of SARs is the product of the nonlinear dynamics of complex systems.

Some errors are minor and should be corrected. For example, in Figure 1 the abbreviations of the studied outcrops are indicated but they are not included in the graph. Another example: some species are not italicized (page 6, line 215). On the other hand, some errors that must be corrected are more important. This is the case of part of the Results with indication of bibliographic citations and that correspond to the Discussion.

In general terms, the Methodology should be clearer in some points. The Results must be restructured. The Discussion is largely speculative. The Conclusions are long and, in part, also speculative, without sticking strictly to the results. On the other hand, it is difficult to understand the number of authors and their contributions. Although the study provides interesting information, I think that its contribution to knowledge is not enough to be published in Plants, but it is in other journals.

Author Response

Reviewer 4.

Comments and Suggestions for Authors

In general, the manuscript is well organized and written, although at times it is somewhat convoluted and unclear. The research addresses a topic of interest but is repetitive. In this case the authors use Species-Area Relationships (SAR) to analyze the performance of gypsum outcrops as edaphic islands, differentiating between different degrees of gypsophily. This methodology is not new but has a long tradition in Population and Community Ecology. In this sense, the authors use the most popular postulates from scientific literature.

Response: In this case, the novelty is not the methodology or, more specifically, the Arrhenius equation. What is new is applying it to the case of gypsum outcrops and following an experimental design that is indeed original. The truth is that the Arrhenius model remains fully valid, as demonstrated by many of the references included in the manuscript, as well as the topic of continental islands.

This work can be enriched if more attention is paid to the parameter c, and also the heterogeneity of the habitat, the susceptibility of species to extinction or the effect of the size of the plots on sampling is taken into account.

Response: All the topics suggested are undoubtedly of interest, but they are not closely related to the objectives of this research. As you know, there is a wide group of researchers worldwide dealing with the ecological aspects of plants living in gypsum. There are at least 3 or 4 reviews since Parsons' publication in 1977 on this type of plants, and there is no doubt that much remains to be done. In any case, the size of the sampling plots, for example, conforms to one of the most widely used methodologies in the research of diversity and richness of plant communities. It is true that it has not been employed in the specific case of gypsophilic plants, so a great opportunity to compare these ecosystems with others sampled with the type of plots we have used in other similar areas worldwide has been missed. Although we have not highlighted this contribution of our research, we consider it a strong point of it.

Ultimately, it cannot be ruled out that the origin of SARs is the product of the nonlinear dynamics of complex systems.

Response: Indeed, it cannot be ruled out. In fact, it has been suggested. We do not intend that research on gypsum plants should continue to expand with multiple approaches. We are working on it.

Some errors are minor and should be corrected. For example, in Figure 1 the abbreviations of the studied outcrops are indicated but they are not included in the graph. Another example: some species are not italicized (page 6, line 215). On the other hand, some errors that must be corrected are more important. This is the case of part of the Results with indication of bibliographic citations and that correspond to the Discussion.

Response: All of that has been corrected. Thank you.

In general terms, the Methodology should be clearer in some points. The Results must be restructured. The Discussion is largely speculative. The Conclusions are long and, in part, also speculative, without sticking strictly to the results.

Response: The text has been deeply modified taking into account the observations of all the referees.

On the other hand, it is difficult to understand the number of authors and their contributions. Although the study provides interesting information, I think that its contribution to knowledge is not enough to be published in Plants, but it is in other journals.

Response: Thank you for your insightful comments on our manuscript. As the great Albert Einstein once said, 'Everything should be made as simple as possible, but not simpler.' While we understand your concern regarding the number of authors and their contributions, we believe our collaborative effort enriches the study with diverse perspectives and expertise, making it more comprehensive.

Response: As for the perceived contribution to knowledge, we are reminded of another famous quote, this time from Sir Winston Churchill: 'Success is not final, failure is not fatal: It is the courage to continue that counts.' We appreciate your feedback and will strive to enhance the knowledge of the plants living on gypsum beyond the SAR model.

Round 2

Reviewer 1 Report

Comments and Suggestions for Authors

The authors have addressed some of the issues raised in my previous review but there remain others which have not been solved so far. In fact, they provide answer to the reviewer in the cover letter but I would rather prefer that they justify their methodological choices to the readers within the manuscript itself.

For example, as a reader, I still miss in this version an explicit definition of some of the concepts that appear in the Introduction such as "the degree of insularity " or "the insularity effect".

With respect to the statistical models and test employed, I suggested including covariates  to partial out effects other than outcrop area that could affect species richness, as the simple Arrhenius-type models assume that environmental conditions in the "islands" are similar (even MacArthur and Wilson 1967  mention that), which may not be the case for these edaphic islands. The authors argue that as they make pair-wise comparisons the effects of covariates would cancel out. This is not true because different "functional" or "taxonomic" groups" may respond differently to those covariates (i.e., there may be statistical interactions between group type and any of the covariates) but, even if those interactions were not significant, excluding covariates would assign all the effect on species richness (and therefore, the variation of the z coefficient) to outcrop area, when it may be due to other causes. I respect the author's choice but I suspect that some readers wouldn't be completely satisfied.

Author Response

Reviewer 1

Comments and Suggestions for Authors

The authors have addressed some of the issues raised in my previous review but there remain others which have not been solved so far. In fact, they provide answer to the reviewer in the cover letter but I would rather prefer that they justify their methodological choices to the readers within the manuscript itself.

Response: Considering that, without a doubt, the section dedicated to materials and methods has undergone the most significant number of changes in the manuscript, we are not entirely sure what the reviewer is referring to in this case. Perhaps they overlooked the fact that in the introduction, we also refer to an article (reference number 26) that coincides in many methodological aspects with ours, as it distinguishes, for example, different taxonomic groups. In this new version, we have endeavored to make this approach even more evident.

Response: Regarding the comments we included in the cover letter but not in the main text of the article, perhaps you are referring to the issue of covariates? If so, we don't think it's appropriate to confuse the reader by explaining something that we have left out of our analyses. By the way, we are not the only ones to make this "exception." You can see examples in many of the articles we cite, especially those associated with references 26 and 46. Both have very similar approaches to what we have done here. In the Materials and Methods, we have included a sentence to make this even clearer.

For example, as a reader, I still miss in this version an explicit definition of some of the concepts that appear in the Introduction such as "the degree of insularity " or "the insularity effect".

Response: On the contrary to what is suggested, we did take this observation into account. In fact, we responded to it by pointing out that, associated with the point where the "degree of insularity" is mentioned, a reference was provided that defines this concept. Here's our response again to that more than apt suggestion: “That's also a good observation. Few authors define it when addressing SAR, but MENDEZ-CASTRO ET AL. do. So, we have placed this reference, of great value to our research, in the appropriate place in the text. Thank you”.

With respect to the statistical models and test employed, I suggested including covariates  to partial out effects other than outcrop area that could affect species richness, as the simple Arrhenius-type models assume that environmental conditions in the "islands" are similar (even MacArthur and Wilson 1967  mention that), which may not be the case for these edaphic islands. The authors argue that as they make pair-wise comparisons the effects of covariates would cancel out. This is not true because different "functional" or "taxonomic" groups" may respond differently to those covariates (i.e., there may be statistical interactions between group type and any of the covariates) but, even if those interactions were not significant, excluding covariates would assign all the effect on species richness (and therefore, the variation of the z coefficient) to outcrop area, when it may be due to other causes. I respect the author's choice but I suspect that some readers wouldn't be completely satisfied.

Response: Thank you very much for respecting our choice, which, as you rightly point out, is based on reasoning. We consider it difficult to justify the use of covariates in the analyses when all species, regardless of their group, are located in the same sampling plots. We are aware that not only can different considered groups respond differently to parameters other than the surface of the outcrop, but the same could be said for each individually considered species. However, it's important to note that there are also randomly created groups that ensure a very robust treatment of the data, in our opinion. Anyway, thanks for your helpful comments.

Reviewer 4 Report

Comments and Suggestions for Authors

The manuscript can be published in the current version.

Author Response

Reviewer 4

Comments and Suggestions for Authors

The manuscript can be published in the current version.

Response: Thank you very much your helpful comments.
